# Spatial Distribution Evolution of Residual Stress and Microstructure in Laser-Peen-Formed Plates

**DOI:** 10.3390/ma13163612

**Published:** 2020-08-15

**Authors:** Zheng Zhang, Wen Huang, Guoxin Lu, Yongkang Zhang

**Affiliations:** 1Institute of Machinery and Manufacturing Technology, China Academy of Engineering Physics, Mianyang 621000, China; CAEP-ZhengZhang@hotmail.com; 2School of Electro-Mechanical Engineering, Guangdong University of Technology, Guangzhou 510000, China; ykzhang@gdut.edu.cn; 3School of Materials Science and Engineering, Shandong University, Jinan 250000, China; luguoxin@live.cn

**Keywords:** laser peen forming, residual stress, hardness, microstructure, Al 7055

## Abstract

Residual stress in structural components is crucial as it affects both service performance and safety. To investigate the evolution of residual stress in a laser-peen-formed panel, this study adopted two plate samples of thickness 3 and 9 mm instead of the conventional Almen strip. The two plates were peened with an identical energy density of 10.99 GW/cm^2^. The residual stress across the entire section was determined using a slitting method, and near-surface stress was then verified by X-ray diffraction. Furthermore, cross-sectional variation in hardness and microstructure were characterized to understand the residual stress evolution. The experimental results showed that different thicknesses resulted in distinct spatial distributions of residual stress. The 3-mm plate demonstrated a shallow (0.5 mm) and lower compressive stress magnitude (−270 MPa) compared with a deeper (1 mm) and higher compressive stress (−490 MPa) in the 9-mm plate. Further analysis revealed that the deformation compatibility during the forming process inevitably leads to a stress compensation effect on the peened side. The decrease in the depth and magnitude of the compressive residual stress in the thin plate was mainly attributed to low stiffness and large deflection.

## 1. Introduction

High-energy and nanosecond-pulse width laser irradiation can induce high-pressure shockwaves, which provide unique advantages in material processing, such as ultrahigh strain rate (10^6^/s), deep compressive stress (1–2 mm), and low work hardening [1]. Depending on the target areas (regional or complete) and usage of the induced deformation (which may be eliminated or utilized), researchers have developed new techniques of laser shock peening (LSP) [2] for critical region strengthening and laser peen forming (LPF) [3,4] in metal sheet contour forming. As opposed to LSP treatment, which is intended to minimize undesirable deflection [5,6], LPF is used to precisely control the deformed shape while maintaining a favorable compressive stress. Compared with traditional shot peen forming, laser beam facilitates a better accessibility and flexibility during the treatment of complex geometrical components [1,7]. Therefore, LPF is fast emerging as an advanced technique for metal sheet forming, particularly in the aviation industry.

Target geometry plays an important role in the evolution of residual stress and deformation contours during the peen forming process. First, the geometrical thickness affects shockwave propagation and plastic strain distribution [8], which are causally related to the generation of a compressive stress field. For thin metal foil/film (micron-level thickness), the shockwave can easily penetrate the entire cross section and result in completely plastic deformation. A mold/die is typically adopted to hold the target and imprint a predetermined 3D structure. This process is also proposed as laser dynamic forming [9] or laser shock imprinting [10]. For thin sheet/plate (thickness at the millimeter level), the shockwave propagates, reflects, and dissipates in the component, resulting in distinct plastic strain distributions depending on the thickness. Under peening, a thin section may encounter the superposition of incident and reflected shockwaves, which could influence the compressive stress on the peened side [11]. A thick section tends to demonstrate more obvious plastic deformation, because thickness can provide enough of a constraint to the shockwave propagation. Second, geometric thickness has a profound effect on the forming deformation, because bending stiffness is strongly related to thickness. For a given LPF condition, forming deformation is negatively correlated to thickness [12]. Moreover, the LPF technique is based on the deformation compatibility effect caused by laser-induced near-surface compressive stress. Therefore, the residual stress evolving in the target component is the key factor to control forming deformation.

Aerospace applications require aircraft wing panels of variable thicknesses as well as precisely designed curvatures in order to maintain good resistance to fracture and stress crack corrosion. To evaluate the peening intensity and residual stress magnitude prior to forming the panel, parametric experiments are typically performed on Almen strips. Surface residual stress and arc height are measured as reference parameters in the panel-forming process. However, the residual stress state of an Almen strip is expectedly different from that of a formed panel. This is because the limited thickness and geometrical parameters of the Almen strip differ from those of a completely peened panel. Furthermore, conventional surface and near-surface stress measurement techniques such as X-ray diffraction (XRD) [13,14,15] and incremental hole drilling [13,14,15,16] may not be sufficient to provide adequate through-thickness stress data for quality control in the forming process due to their limited depth penetration. Given that the knowledge of residual stress in the peen-formed component, not just the test sample, is critical, the evolution of the residual stress field needs to be better understood. In addition to the test sample prior to LPF, plate coupons with different thicknesses undergoing complete peening appear to be more representative and useful for engineering applications.

The objective of this research was to investigate the evolution of through-thickness residual stress in a completely peen-formed plate. Unlike traditional parameter tests using Almen strips, this study adopted two large sheets, one 3-mm thick and the other 9-mm thick, to provide a better comprehensive reference on residual stress and forming deformation. The residual stress profile across the entire section was determined using a slitting method, and the near-surface stress was then verified by XRD. Furthermore, cross-sectional characterization of the microstructures of the peened plates was performed to gain a better understanding about the residual stress evolution. Finally, the correlation between the deformation compatibility and residual stress distribution in the two plates is discussed in detail.

## 2. Materials and Methods

### 2.1. Materials and LSP Treatment

The aluminum alloy 7055-T7751 was specifically developed for use in compression-dominated structures, such as aircraft upper wing panels, horizontal stabilizers, and keel beams. Therefore, a 25-mm-thick AA7055-T7751 pre-stretched plate (Alcoa, Pittsburgh, PA, USA) was selected for this study. These 7000 series aluminum alloys cannot be heat treated (annealed) to relieve stress while maintaining the original mechanical properties. Typically, cold works such as stretching and compression are adopted to overcome this problem. The number T-51 indicates that a 1.5–3% uniaxial stretching was applied in the rolling direction to relieve residual stress.

Two rectangular sheets were extracted from a base plate using wire electrical discharge machining (EDM) to a planar size of 300 mm × 100 mm. The minimum and maximum thickness of typical wingtips were selected to study the corresponding forming properties. Thus, the thicknesses of the two sheets were 3 and 9 mm, respectively. Note that the two sheets inherit the same orientation as the base material—that is, 300 mm length in the rolling direction (RD) and 100 mm width in the transverse direction (TD). Then, a 3-axis machine center was employed to remove the material layer-by-layer to a thickness of 3 mm and 9 mm for the two sheets. A specially designed vacuum fixture was used for holding the sheets based on air pressure difference between the top and bottom surfaces. A finish milling operation was conducted on the sheet surface to achieve a roughness of *Ra* = 1 μm prior to the LSP treatment.

The residual stress magnitude is primarily affected by the mechanical properties of the base material. Anisotropy and orientation might result in deviation in the biaxial in-plane residual stress [17,18]. Thus, it is necessary to evaluate the mechanical properties prior to forming. Two samples extracted from the RD and TD of the base plate were used in the tensile test. Table 1 shows a comparison of the mechanical properties of the two samples. No significant variations in the yield strength and ultimate tensile strength were detected for different orientations of the base material.

LSP treatment was performed on a Q-switched Nd:YAG laser system (SGR Extra-08, Beamtech, Beijing, China). The two sheets were completely peened with identical parameters, and they could be deformed freely throughout the forming process. Black tape (471, 3M, Saint Paul, MN, USA) was used as the opaque overlayer and a 1.5-mm laminar flow (water; flow measured using an ACR-HNDS100 Laser Focus Displacement Meter, Schmitt Industries, Inc., Portland, OR, USA) was used as the transparent overlayer. The LSP process parameters are shown in Table 2.

### 2.2. Determination of Residual Stress

Since the residual stress state in the peen-formed plate varies primarily in the depth direction and satisfies the plane stress assumption, the slitting technique [19] was selected to quantify the biaxial in-plane residual stress due to its ability for full depth penetration. To ensure accurate measurements, the Los Alamos National Laboratory Report [20] and practical instructions [21] were carefully followed. A wire EDM with a 0.2-mm-diameter brass wire was employed to cut the slot (0.25 mm).

For the 3-mm sheet, 50-μm incremental cuts were made throughout the measurement process, resulting in a total of 54 cuts to a depth of 2.7 mm (90% of the thickness). For the 9-mm sheet, 100-μm incremental cuts were made to an initial depth of 2 mm, and a 200-μm incremental cut was employed for the remaining depth. The resulting depth was 8.6 mm (95% of the thickness), with a total of 53 cuts. A strain gauge (125LW, Vishay, Malvern, PA, USA) was used to measure the released strain. The gauge was attached to the back face of the sheet and covered with a waterproof coating (3145RTV, Dow Corning, Midland, MI, USA). For each LPF sheet, two independent samples were extracted from the RD and TD orientations to determine the in-plane stress components.

Note that the initial bulk stress within the two sheets was assumed to be negligible due to its low magnitude. Preliminary testing revealed that the bulk residual stress in the pre-stretched AA7055-T7751 was in the range of −25~30 MPa (Figure 1a), similar to that of AA7050-T7451 [22] but with a magnitude that is about 10 MPa greater. After the extraction and milling processes, the material removal resulted in the relaxation of residual stress, and thus the two plate samples inherited a relatively low initial stress condition (<30 MPa).

Additionally, laboratory X-ray diffraction (XL-640 ST, X-ray stress analyzer, Stress Technologies, Handan, China) was performed to characterize the in-plane near-surface stresses. The {3 1 1} crystallographic plane was detected using a Cr Kα tube with a diffraction angle 2θ = 133–144°. The depth profile was acquired using incremental electropolishing and subsequent XRD measurement. The depth was measured using a micrometer, and the results were derived from the average of five repeated measurements.

In addition to the milling-induced stress of the plate samples, the measurement indicated that the milling-affected layer extended up to 80~100 μm (Figure 1b). Given that identical milling operations were performed on the two plates and that the peening after milling was conducted conventionally, the initial stress condition was assumed to be the same in this study.

### 2.3. Surface Roughness, Hardness, and Microstructure Measurements

First, the surface qualities of the two sheets were measured using a roughness tester (MarSurf UD 130, Mahr, Esslingen, Germany). Then, a wire EDM was used to cut samples from the base material and LPF sheets in both the RD and TD. The test samples were manually polished to obtain a smooth cross section for measurement. A microhardness tester (FALCON 5000, Innovatest, Maastricht, Netherlands) was used to measure the hardness variations along the depth direction in both the RD and TD orientations. The indentation load was 100 gf and was held for 15 s. For each measurement, a line of 60 indents was made with 50-μm spacings, which started from the LSP-treated surface and extended to a depth of 3 mm. Finally, the microstructures of the peened sheets were explored with an optical microscope (DM2700 M, Leica, Wetzlar, Germany) and electron backscattered diffraction (EBSD) (Symmetry, Oxford Instruments, Abingdon, UK) to characterize the shockwave-induced evolution.

## 3. Results and Discussion

### 3.1. Spatial Distribution of Residual Stress in Completely Peen-Formed Plates

Figure 2a,b shows the through-thickness distribution of the residual stress in the 3- and 9-mm LPF sheets, respectively. Note that the range of bulk residual stress in the base plate is also plotted for comparison (−25/+30 MPa baseline). The slitting results clearly indicate that both sheets exhibited an approximately equal biaxial stress state. The through-thickness distributions of the RD and TD residual stresses were similar; that is, the tension in the center was balanced by the compression on both the treated and untreated sides. The residual stress across the entire section satisfied the overall equilibrium. It can also be seen that surface stresses transferred from tension (Figure 1a) to compression. The XRD and slitting results were in good agreement for the near-surface region, which validated the observation that the in-plane stress components in RD and TD were roughly of the same magnitude.

However, significant differences in the compressive stress distribution on the peened side were observed between the two sheets. The 9-mm sheet exhibited an obviously higher stress amplitude (approximately 490 MPa, 88% yield strength) than the 3-mm sheet (about 270 MPa, 49% yield strength). In addition to the depth of the compressive stress, the 3-mm sheet exhibited a 0.5-mm compression layer, whereas the 9-mm sheet exhibited a 1.0-mm compression layer, although identical processing was applied to both sheets. This implies that the shockwave-induced plastic deformation was weakened by the reflected wave in the 3-mm plate due to lack of constraint. In contrast, the 9-mm thick section could provide sufficient volume for the shockwave propagation and dissipation, resulting in a high level of compression. To confirm this hypothesis, a direct verification of shockwave-induced plastic deformation in the two LPF sheets was needed.

### 3.2. Surface Roughness, Hardness, and Microstructure

High-pressure-induced plastic deformation is often accompanied by obvious microstructure evolution and work hardening [7,8]. Accordingly, surface roughness and cross-sectional hardness were selected for a quantitative comparison.

Figure 3a,b shows the surface roughness of two sheets before and after LSP treatment, respectively. Prior to laser peening, it can be clearly seen that both the milled surfaces exhibited 1 μm roughness. After laser peening, the surfaces received multiple impacts and the resulting plastic deformation was in the range of ±4 μm. Compared with other laser-peened aluminum [13,14], the deformation depth was shallow. This can be attributed to the greater hardness of AA7055 [23]. The surface contours of the two sheets were similar and no significant difference was found. Apparently, identical shockwave pressure was applied on the surfaces. Consequently, the near-surface materials were plastically deformed with equal amplitude.

The cross-sectional hardnesses of the untreated material, the 3-mm, and the 9-mm LPF sheets are shown in the Figure 4. Prior to LSP treatment, the base material demonstrated a relatively good consistency in both directions, with an average hardness of 178.6104 HV and 180.5665 HV in the RD and TD, respectively (Figure 4a). After laser peening, a slight increase was observed at the peening-affected depth (1~1500 μm) in both 3- and 9-mm sheets. The hardness approximately increased by 5.56~7.78% (average 190~194 HV) as compared with the untreated condition in Figure 4a. These observations are consistent with the minor enhancement in hardness often observed in aluminum samples under single impact [13,14]. Additionally, no significant difference in the hardness variation was found between the two sheets. This may imply that the effect of shockwaves on the hardness enhancement was similar in the 3- and 9-mm sheets.

Preliminary study has revealed that the elongated grains and texture in the base material can mainly be attributed to the rolling and stretching processes. The spatial distribution of grain size was not uniform (Figure 5). The cross-sectional microstructures of the 3- and 9-mm LPF sheets after laser peening are shown in Figure 6. The RD and TD orientations were still dominated by an elongated grain. No significant grain refinement or gradient were observed on the peened surfaces of the two sheets, although obvious plastic stress was generated (Figure 2). The EBSD Euler angle orientation maps in Figure 7 also display similar grain distribution patterns. Moreover, no phase changes were detected on the peened surfaces by the EBSD. These nonsignificant evolutions might imply that the texture caused by cold stretching was superior to that created by the high-pressure effect. This is because T-7751 temper plates typically possess high mechanical strength, and the uniaxial stretching results in a stable texture [22]. The slight increase in hardness (Figure 4) also proved this hypothesis. In addition, other studies [13,14] indicated that the hardness increase and microstructure evolution of pre-stretched aluminum are not sensitive to a single peening.

### 3.3. Evolution of Residual Stress and Deformation Compatibility

The surface quality, hardness, and microstructure analysis indicated that there was no direct evidence of relation to the plastic deformation differences between the two peen-formed sheets. The distinct distributions of compressive stress in the two plates of different thicknesses may thus be attributed to another factor.

Figure 8a shows a substantial difference in the forming deformation of the 3- and 9-mm sheets subjected to identical peening conditions. This result is attributed to the fact that the deformation magnitude is primarily influenced by the structural stiffness (i.e., thickness) and that deflection and stiffness are negatively correlated [24]. Apparently, the 3-mm sheet inherited a lower stiffness and demonstrated a large deflection (37.1910 mm), whereas the 9-mm sheet inherited a higher stiffness and showed a small distortion (4.7080 mm). In the case of the large deflection, the bending strain and stress showed an obvious non-linearity in the thin section (Figure 8b). In contrast, the bending strain and stress demonstrated an approximately linear distribution in the thick section due to the small deformation (Figure 8b). This analysis is supported by Figure 2, which shows two distinct stress patterns across the section (from tension to the un-peened side). The 3- and 9-mm sheets demonstrated a nonlinear and an approximately linear stress pattern, respectively.

Essentially, the final state of the residual stress (*σ_final_*) in the LPF sheet is a superposition of the peening-induced elastoplastic stress (*σ_LSP_*) to the affected depth and the elastic bending stress (*σ_bend_*) across the thickness (Figure 8b). It should be noted that compressive stress on the peened side must inevitably be compensated by cross-sectional bending stress due to stress equilibrium and deformation compatibility. Consequently, the thick sheet inherited small deformation and low bending stress, resulting in lower compensation for compressive stress on the peened side. The thick section could provide adequate constraints to maintain a deep depth and high magnitude of compressive stress on the peened side. In contrast, the thin sheet inherited large deformation and high bending stress, resulting in compressive stress being compensated to a greater extent on the peened side. The thin sheet lacked enough constraints, resulting in an increased tensile region to balance the high compression on the treated surface (i.e., a decrease in the compressive stress depth).

In summary, although LSP introduced obvious residual stress and forming deformation, no significant difference in plastic deformation was observed between the two peen-formed sheets. The through-thickness distribution of the residual stress was evidently influenced by the deformation compatibilities of the different plate thicknesses. When dealing with thin-walled structures, the depth and amplitude of residual stress might differ with variations in thickness due to stress equilibrium. Therefore, extra attention is required in such cases.

## 4. Conclusions

1. Despite identical laser peening conditions (10.99 GW/cm^2^), sheets of different thicknesses resulted in distinct residual stress distributions. The thin sheet was prone to inheriting a shallow (0.5 mm) and low level of compressive residual stress (−270 MPa, 49% yield strength), whereas the thick sheet could maintain a deep compression layer (1 mm) of high magnitude (−490 MPa, 88% yield strength).

2. The structural deformation compatibility inevitably led to a stress compensation effect on the peened side. The decrease in the depth and magnitude of the compressive residual stress in the thin plate was mainly attributed to low stiffness.

3. In this study, the two plate samples effectively served as solid experimental references for residual stress and forming deformation. For engineering applications, it is recommended that plate samples of variable thicknesses be used prior to actual component forming because the presence of residual stress is crucial to safety.

## Figures and Tables

**Figure 1 materials-13-03612-f001:**
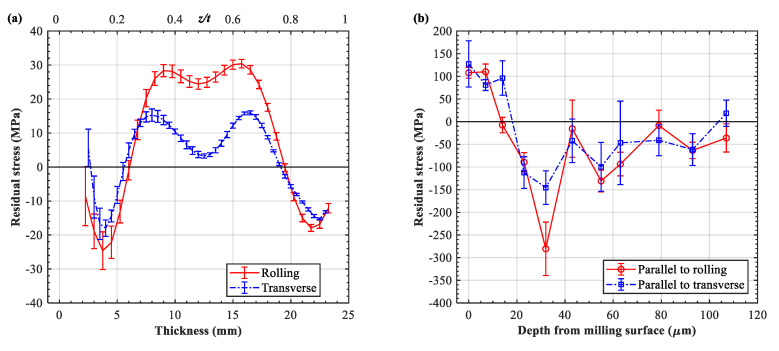
Initial conditions of residual stress of sheet samples: (**a**) bulk residual stress of base plate and (**b**) milling-induced near-surface residual stress.

**Figure 2 materials-13-03612-f002:**
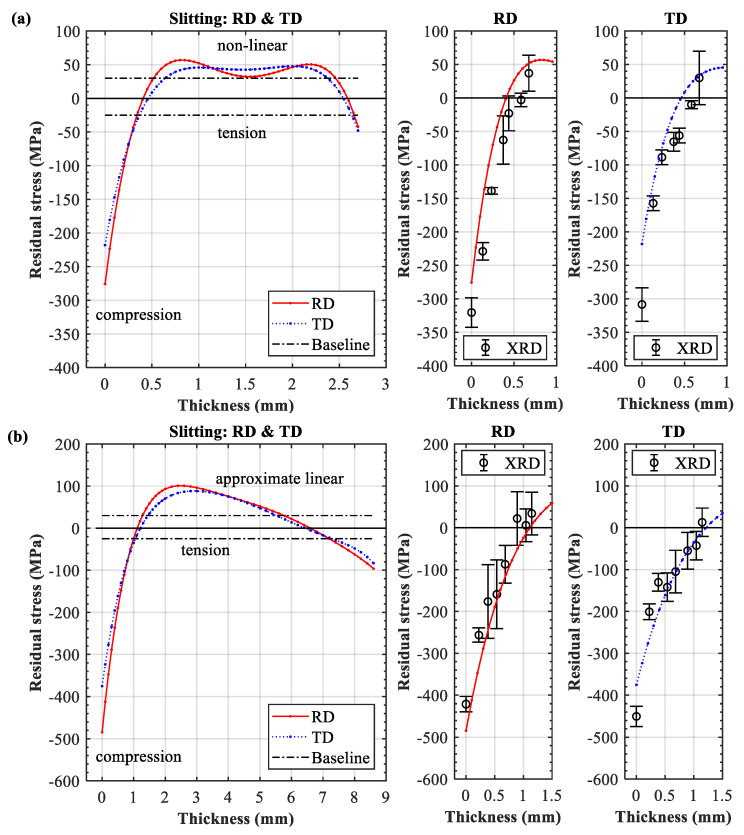
Spatial distributions of residual stress in the sheets formed by laser peen forming (LPF): (**a**) 3-mm sheet and (**b**) 9-mm sheet.

**Figure 3 materials-13-03612-f003:**
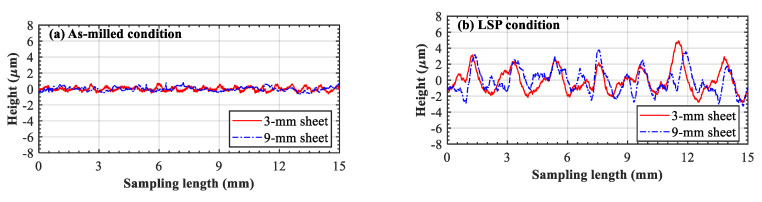
Surface roughness of the two LPF sheets: (**a**) as-milled condition and (**b**) laser-peened condition.

**Figure 4 materials-13-03612-f004:**
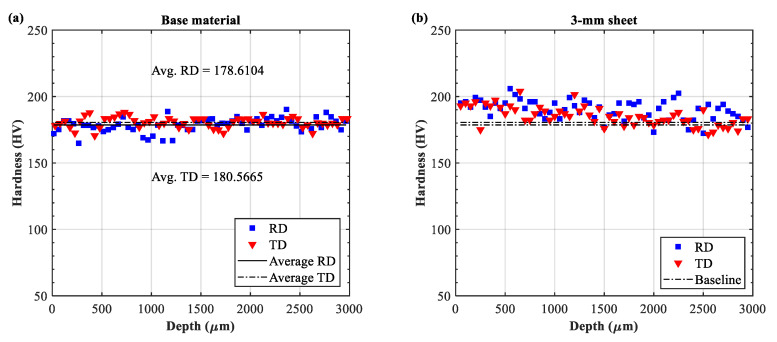
Cross-sectional hardness variation in the base material and the LPF sheets: (**a**) base material, (**b**) 3-mm sheet, and (**c**) 9-mm sheet.

**Figure 5 materials-13-03612-f005:**
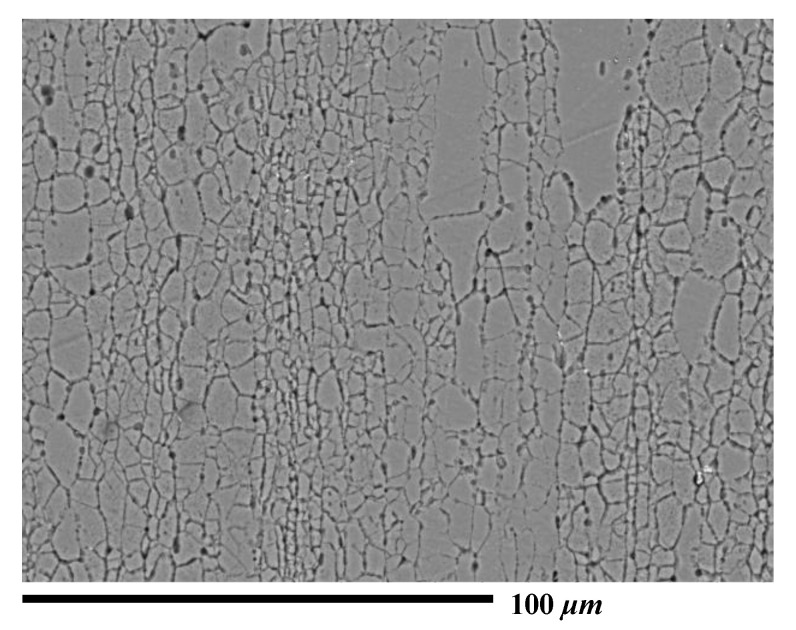
Microstructure of the base material in the rolling direction (untreated condition).

**Figure 6 materials-13-03612-f006:**
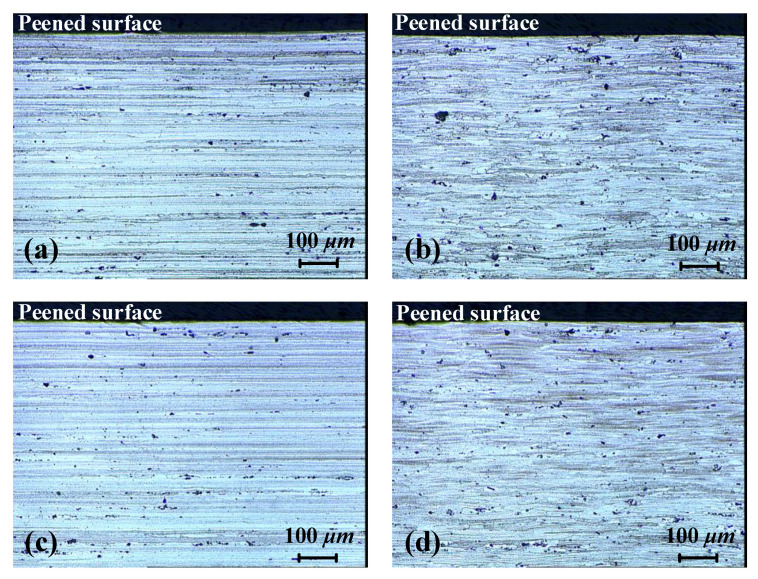
Cross-sectional microstructure of the 3- and 9-mm LPF sheets: (**a**) 3-mm sheet in the rolling direction (RD), (**b**) 3-mm sheet in the transverse direction (TD), (**c**) 9-mm sheet in the RD, and (**d**) 9-mm sheet in the TD.

**Figure 7 materials-13-03612-f007:**
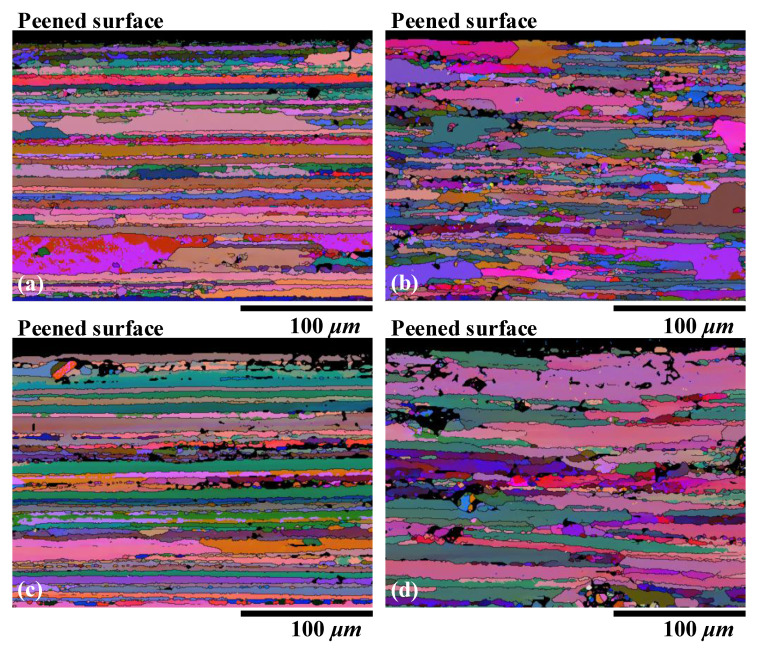
Cross-sectional EBSD Euler angle orientation maps of the 3- and 9-mm LPF sheets: (**a**) 3-mm sheet in the RD, (**b**) 3-mm sheet in the TD, (**c**) 9-mm sheet in the RD, and (**d**) 9-mm sheet in the TD.

**Figure 8 materials-13-03612-f008:**
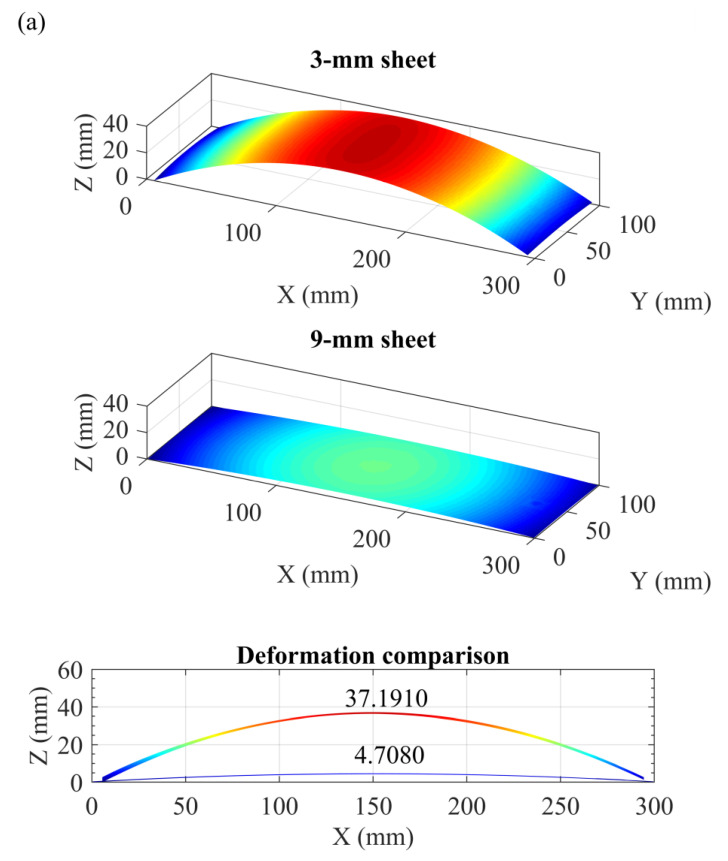
Comparison of forming deformation and evolution of residual stress in the two LPF sheets: (**a**) measured global deformation of the two LPF sheets and (**b**) effect of deformation compatibility on residual stress distribution.

**Table 1 materials-13-03612-t001:** Mechanical properties of test sample AA7055-T7751.

Orientation	Yield Strength (MPa)	Ultimate Tensile Strength (MPa)	Elongation (%)
RD	553.8602	603.6828	12.4742
TD	545.1015	603.7197	12.0130

**Table 2 materials-13-03612-t002:** LSP treatment parameters.

Parameters	Value
Laser wavelength (nm)	1064
Laser pulse energy (J)	7
Laser spot diameter (mm)	2.6
Laser pulse width (ns)	18
Power density (GW/cm^2^)	10.99
Overlapping rate (%)	30
Repetition rate (Hz)	2

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
