# Peer review of "Spatial Distribution Evolution of Residual Stress and Microstructure in Laser-Peen-Formed Plates"

_materials, 2020, doi:10.3390/ma13163612_

Round 1

Reviewer 1 Report

There are several issues in the manuscript that should be addressed before further consideration for publication.

  1. How many samples are used to obtain the values in Table 1? Any statistical analysis conducted to conclude that they are not significantly different?
  2. Any further characterisation of the microstructure at the surface? For example, to identify any phase changes/grain size changes?
  3. Any benchmarking done on the residual stress measurement? For example, discuss on the difference between laser peened surface with different parameters? How about the surface quality, any changes?

Author Response

Thank you very much for your valuable comments.

Reviewer 2 Report

Dear authors,

I am deeply moved to have been able to review your paper every interestingly, but I wish to revise the following comments.

(m) mandatory

(o) optional

[1] (m) p. 2 Section 2.1:How did you process the 25 mm thick plate into 3 mm and 9 mm thick specimens? Please explain this process in detail in the manuscript.

[2] (m) p. 2 Section 2.1:Is the test material rolled or extruded? Please describe it in the text. This depends on whether the LD or TD direction is relative to the rolling direction or the extrusion direction.

[3] (m) p. 5 Section 3.2:It is necessary to discuss how much the hardness of the LSP material increases to the hardness of the base material (untreated material). Therefore, please describe the hardness of the base metal in the text and discuss the increase in hardness.

[4] (m) p. 5 Section 3.2:Work hardening is considered to mean an increase in stress with an increase in the strain (dσ/dε). Therefore, it can be considered that the expression "hardness increase" is appropriate than "work hardening" here.

[5] (m) p. 5 Figure 2:In the caption description in Figure 2, it is considered appropriate to the expression "hardness change" rather than "work hardening".

[6] (m) Please correct the indicated part of the manuscript.

Yours sincerely,

Author Response

(The authors gave the same response as above.)

Reviewer 3 Report

The authors report on the investigation of the distribution of residual stress and hardness of aluminum alloy 7055-T7751 plates after laser peen forming. They propose to use samples other than the traditional Almen strips in order to better simulate the real behaviour of components made for aviation industry, such as aircraft upper wing panels, horizontal stabilizers, and keel beams.

The reviewer suggests to accept the article after minor revisions, as follows:

Line 89: why did the author choose the two thicknesses 3 and 9 mm? which component do they want to replicate?

Line 90: the dimension of Ra is missing.

Lines 93-94: how many replications were performed for each orientation?

Line 105 (Table 2): why did the authors choose such laser parameters (i.e. power density, overlapping, repetition rate)? have they performed preliminary investigations to find the best operational parameters? which is the laser scan speed adopted? this is an imporant parameter since it directly influences the interaction time between the laser and the material and therefore the energy transferred.

Author Response

Thank you for your valuable comments.

Round 2

Reviewer 1 Report

NA

Reviewer 2 Report

Good